# Informed Consent in Mass Vaccination against COVID-19 in Romania: Implications of Bad Management

**DOI:** 10.3390/vaccines10111871

**Published:** 2022-11-05

**Authors:** Sînziana-Elena Bîrsanu, Maria Cristina Plaiasu, Codrut Andrei Nanu

**Affiliations:** 1School of Advanced Studies of the Romanian Academy, 125 Calea Victoriei, 010071 Bucharest, Romania; 2Doctoral School, University of Medicine and Pharmacy of Craiova, 2 Petru Rares Street, 200349 Craiova, Romania; 3Department No. 14 of Orthopedics, Anesthesia and Intensive Care, University of Medicine and Pharmacy “Carol Davila” Bucharest, 37 Dionisie Lupu Street, 020021 Bucharest, Romania

**Keywords:** COVID-19, vaccination campaign, medical liability, informed consent, Romanian physicians

## Abstract

Informing patients and obtaining valid informed consent were significant challenges for the COVID-19 immunization program. In Romania, the authorities issued a strategy for activities regarding vaccination against COVID-19, including the informed consent procedure. The lack of legal preparedness was evident when the medical personnel at the vaccination centers were provided with informed consent forms that did not respect the existing legal requirements. In addition, the protocol for persons seeking vaccination stated that the patient was supposed to receive the informed consent form from the receptionist in order to read and sign it. We analyzed the legal implications and the malpractice litigation risk associated with this practice. Due to essential deficiencies and in the absence of an official enactment of new regulations, we conclude that the vaccination consent process did not comply with the legal requirements. Implications include medical personnel’s legal liability, loss of malpractice insurance coverage, and public mistrust that may have contributed to a low vaccination rate. Given the potential of future pandemics or other health crises, this may be a valuable lesson for developing better legal strategies.

## 1. Introduction

The pandemic prompted the world’s states to adjust to vital changes and weigh the community’s welfare against human rights. The Parliamentary Assembly of the Council of Europe encouraged European nations to allow the general population to make their own vaccination decisions [1], and individuals, with some exceptions [2,3,4], could choose whether to be vaccinated.

Vaccination of a large number of individuals was necessary to establish herd immunity [5,6], but, at the same time, vaccination programs had to be implemented in a way that respected patients’ decisional autonomy [7]. Previous studies reported strategies for improving vaccination rates without compromising patients’ right to consent. It was noted that choice architecture and compensations could increase vaccination acceptance [8], as well as good communication of the herd immunity concept [9].

Despite concerted efforts, obtaining valid informed consent and guaranteeing a shared decision-making process for patients were significant challenges of the vaccination campaign against COVID-19, especially for elderly or incapacitated people [10]. Pandemics unprecedentedly challenge the traditional informed consent approach, with direct opportunities for the patient to express questions. Therefore, the COVID-19 pandemic generated several legal issues, including the possibility of a derogatory regime for obtaining patients’ informed permission [11]. In some countries, the patients were not asked to sign an informed consent to be vaccinated [12].

In Romania, the National Committee for Coordination of Activities regarding Vaccination against COVID-19 (CNCAV) coordinated the vaccination campaign. This national committee was established by the Prime Minister’s Decision no. 385/20.11.2020. The Romanian Government approved the vaccination strategy against COVID-19 through the Government Decision no. 1031/2020, with subsequent amendments and additions, and a national platform dedicated to the vaccination campaign was launched [13].

The Romanian legislation remained unaltered despite calls for a new and distinct legislative framework to clarify the roles and responsibilities of all parties involved in the vaccination against COVID-19 [14]. Romanian law clearly states what type of information physicians must provide to patients. In addition, the law requires the use of a unique informed consent form. Without any derogatory regime for informed consent regarding the vaccination against COVID-19, the medical personnel at the vaccination centers had to comply with the general legal provisions regarding the informed consent process.

During the vaccination campaign, the medical personnel were provided, inexplicably, with different and illegal informed consent forms to be used [15]. These forms were published on the abovementioned national platform. Two versions of this document are available, one for adults and one for minors.

Patients’ informed consent documents used during the vaccination campaign against COVID-19 differ significantly from the ones required by law, as they exclude essential information. The protocol for immunization against COVID-19 required the patients to read and then sign the agreements. Eight million patients were vaccinated after signing these forms.

We aim to explore the legal implications and the malpractice litigation risk linked to this practice. We examined the regulations governing the medical practice, how Romanian courts handle malpractice cases in which medical procedures were performed without the patient’s consent, and the effects on malpractice insurance coverage agreements.

## 2. Romanian Legal Framework Regarding Patient’s Informed Consent

Under the Romanian laws no. 95/2006 and no. 46/2003, a patient’s written consent is mandatory. The law requires physicians to inform patients and obtain their written consent prior to any medical procedure.

### 2.1. How Informed the Informed Consent Should Be?

Consistent with the Oviedo Convention and European legislation, the Romanian medical legislation respects the patient’s right to be informed about essential elements before consenting to any medical intervention, vaccination included. By law, the mandatory information must contain the elements stated in Table 1.

In the case of COVID-19 vaccination, the Canadian informed consent form included a similar checklist [16]. At the same time, Italy opted for full disclosure [17] and provided patients with an appendix of data on the vaccine, risks included [18]. The United Kingdom determined that cover letters should be issued to individuals who were able to consent and contain information about risks and website links. Patients ultimately received individualized vaccine information and signed a consent form [19].

Furthermore, informed consent is more than a signature on a paper [20]. Obtaining a patient’s informed consent is not simply a routine checking of a legal form [21,22] but a process through which individual values and freedom are expressed. Consent is essential to the therapeutic relationship of trust between the physician and the patient, and the informed patient has the freedom to select and refuse therapy [23]. Romanian jurisprudence acknowledges the importance of a thorough explanation of the medical act and its risks and consequences: “the law requires the doctor to inform the patient about his health status, but this information requires not only a reading of the test result but also an interpretation of this result from the doctor’s perspective” [24].

The law also imposes a series of rules to ensure the accuracy of the informed consent process, as outlined in Table 2.

### 2.2. Informed Consent—A Signature on a Legal Document

Informed consent should be secured through the patient’s signature on a mandated form model in which the patient or the medical care personnel accurately records the information provided [25]. The lawmaker issued this unique informed consent form for use in Romanian medical practice, per the Health Ministry’s Order no 1411/2016 (Appendix A).

All medical personnel are supposed to use this document to acquire the signed informed consent. This document meets all of the aforementioned legal standards.

## 3. Assessing Legal Validity of the Patient’s Informed Consent Used during the Romanian Vaccination Campaign

At the beginning of the Romanian vaccination campaign, the National Committee for Coordination of Activities Regarding Vaccination Against COVID-19 published a twenty-six-page vaccine methodology [26]. Only one sentence is dedicated to patient information, which refers to medical personnel being taught to advise patients about potential adverse effects and what to do in this case.

Additionally, a new vaccination informed consent form was released on the vaccination platform (Appendix B). This COVID-19 vaccination consent includes:
Information related to the patient’s personal data;A pre-filled brief description of the medical act (vaccination);An acknowledgment statement.

The acknowledgment refers only to the following: (i) providing accurate information to the medical staff, (ii) understanding the information received, and (iii) expressing the patient’s decision to accept the vaccine.

We argue that the informed consent form used during the vaccination campaign has no legal standing under procedural or substantive law. We assert that the informed consent form distributed at the vaccination centers had no legal base for the following reasons:
There was no law, order, or other regulation issuing this document. We searched the legal database accessible through legislation software (Indaco Lege5 and Sintact) and Google. Nevertheless, we could not identify any official act adopting this informed consent form used during the vaccination campaign. Therefore, we consider that the vaccination informed consent form made available on the vaccination web platform did not have legal grounds.According to the hierarchic system of laws general principle [27] and Romanian law no. 24/2000, any amendment of a normative act in force is permitted only by an act of higher or equal rank. Consequently, at least a Ministry Order was required for the new informed consent to be legitimate. There is one official document related to the vaccination campaign strategy, Health Ministry Order no. 2171/2020, for establishing norms regarding the authorization, organization, and operation of vaccination centers against COVID-19. This order does not mention any derogation regarding the patient’s informed consent, but it simply states the necessity for the patient to sign an informed consent form. The National Committee for Coordination of Activities Regarding Vaccination Against COVID-19 was not legally authorized to amend law no. 95/2006 on health reform and order no. 1411/2016 of the Ministry of Health and, therefore, to alter the informed consent form. 

Nevertheless, no authority asserts ownership of the informed consent form published on the official vaccination platform.

Furthermore, according to the legal procedure for vaccination against COVID-19 and the protocol for persons seeking vaccination approved by Ministry Order no. 2171/2020, the patient traffic flow is divided into five steps (Figure 1).

In the second step of the protocol, it is stated that the receptionist would deliver the informed consent form to the patient. In the next phase, the patient should read and sign the document in case of agreement. The most critical missing element from the vaccination protocol is providing the patient with information on the benefits and repercussions of the immunization, potential hazards, plausible treatment options, and risks associated with the absence of the immunization. The official Ministry Order contains no phase at which the legal process of informing the patient required by Romanian and European legislation should occur.

The law mandates that information be supplied directly by the physician and tailored to each patient. Therefore, the patients were entitled to receive not only general information about the inherent dangers of the vaccines but also about risks specific to each patient [28]. Under no circumstances should media or online sources substitute for physicians’ medical information.

The entire procedure clearly shows that in Romania, even at the highest levels, legislation regarding patients’ right to informed consent was completely ignored during the vaccination campaign.

This was not the case in other countries where physicians were instructed to adhere strictly to the law. France encouraged physicians to inform the patients and honor their free and explicit choice [29]. Similarly, the Canadian vaccination guide required the disclosure of minor and severe adverse effects of the vaccine before administration [30].

Due to an essential deficiency and in the absence of an official enactment of new regulations, we consider that the vaccination consent form should not have been used in any medical facility.

## 4. Legal Implications of Deficient Informed Consent during the COVID-19 Vaccination Campaign

Delivering no information or insufficient information to patients before immunization and using an illegal informed consent form has increased the risk of physicians’ and medical personnel’s liability during the Romanian vaccination campaign.

According to Romanian medical legislation, medical malpractice could result in physicians’ civil liability. As per article 653 paragraph 1 of law no.95/2006 on health reform, malpractice is professional misconduct that occurs during the rendering of medical services and causes patient injury, leading to the civil liability of medical personnel.

When defining medical personnel’s civil liability, article 653 paragraph 3 of law no. 95/2006 states that the medical personnel will also be liable for any damages and injuries resulting from failure to comply with the regulations regarding confidentiality, informed consent, and the obligation to provide healthcare.

In the case of the COVID-19 immunization program, the physicians’ liability was linked to their responsibility to inform the patient [31]. Providing the required information after assessing the patient’s competence [32] and securing the patient’s approval fulfills the physician’s duty to acquire informed consent. The absence of any mandatory piece of information invalidates the patient’s consent. Although physicians are not required to confirm that the patient has comprehended information, they must provide all the mandatory data.

In the event of a claim, the patient’s signature on the consent form will serve as evidence. Therefore, physicians should document what was discussed with the patient and all other components of the informed consent [33].

Romanian courts of law also emphasize the legal ramifications of insufficient informed consent. According to the court, clinicians who neglect to inform patients about the nature of the medical investigation and its consequences may be held liable.

For example, a physician was sentenced to compensatory damages of about 47,000 euros for failing to inform one of his patients [34]. The physician stated in his defense that he lacked time to notify the patient due to his high patient load. The court decided that physicians have two legal duties: informing the patient about the nature and consequences of the medical investigation and obtaining the patient’s written consent. Organizational issues cannot justify violating a patient’s fundamental right.

The written form of a patient’s consent is mandatory to prove precisely how the patient’s rights were respected. A contrary interpretation would permit medical personnel to take advantage of the patient’s condition of vulnerability and invoke in a lawsuit the existence of verbal permission, which would be easily demonstrated in the particular context, thus exempting the physician from civil culpability.

According to the court’s decision, the states that generally accepted complications and risks of investigation and treatment methods might exonerate doctors from their liability, but this cause cannot be accepted as long as the patient was not aware of the intervention’s risks and the doctor did not request their consent.

Physicians involved in the Romanian vaccination campaign cannot prove they successfully presented the patients with accurate and complete vaccination information. On one hand, there is no evidence of the information presented to the patient or even that the medical personnel presented all the legally required information. The patient’s signature for vaccination only proves that the patients were not physically constrained to vaccinate. On the other hand, the fact that a patient obtained vaccination information from the internet or the media cannot be used as a defense by any authority or physician.

In the absence of valid informed consent, medical personnel are responsible for any harm the patient may suffer, even if no medical error occurs. In this case, medical responsibility is either based on the concept of “loss of chance” for the patient to choose the best treatment or “lack of psychological preparation” for confronting the risk of injury [35].

Furthermore, some authors argue that physicians may face criminal charges in cases of improper informed consent [36]; the intervention is regarded as an assault on the human body [37].

## 5. Other Implications

### 5.1. Medical Malpractice Insurance Agreements

One significant consequence of using non-valid informed consent is the non-operation of professional civil liability insurance agreements. According to medical law, the physician is directly accountable for compensating losses produced by an unconsented-to medical procedure.

Lack of patient informed consent represents an exclusion cause in almost all malpractice insurance agreements, as exemplified in Table 3. Supposing that there is no such exclusion in the agreement, the company would pay but has the right to recoup any payments from the physicians who provided medical assistance without informed consent, according to law no. 95/2006.

Furthermore, working at a hospital does not absolve physicians of their responsibility to cover the prejudice. Romania’s medical liability concept is based on separate hospital and physician accountability. According to the law, medical personnel are legally accountable for any damages caused by negligence or failure to comply with regulations governing patient informed consent. The hospital is liable for any damages created by the working environment (nosocomial infections, medical equipment with known defects, expired medicines, or sanitary materials).

In the issue of vaccination, if there is an employment agreement and the physician’s misconduct occurs while executing hospital-assigned tasks, the hospital may be held civilly liable [38,39,40]. However, since the hospital’s insurance company would not pay the compensation [41], a regressive action against the physician is required by law to recover the payments.

### 5.2. Low Vaccination Rate

By week 42, 2022, 42% of Romania’s population had received at least one dose of a vaccine, compared to an average cumulative uptake of 75% across 30 European nations [42]. Other European nations share the same problem, but there is no single strategy that should be implemented to combat vaccine reluctance [43]. There are many reasons for COVID-19 vaccine hesitancy, such as concerns about adverse events, a lack of adequate information, and a failure of vaccination education, publicity, and popularization [44]. Removal of any reference of potential complications [45] from the informed consent raises public concern regarding the vaccines’ side effects. Another reason for COVID-19 vaccine hesitancy is the lack of communication from trusted providers [44]. The most trustworthy providers are and should be vaccination-campaign-involved physicians with access to the mechanism for getting thorough informed consent.

Using a different consent form than the one patients were accustomed to signing (as per Appendix A) did not aid in creating confidence in vaccination campaigns, as increasing confidence requires providing patients with transparent medical information [46]. Additionally, giving personalized communication from reputable sources such as healthcare providers is one potential way to promote vaccine uptake [44]. We believe that the informed consent phase is the optimum time to provide information.

Moreover, the research emphasizes the significance of personalized vaccination knowledge and education, and the correlation between vaccination acceptance and a level of trust in physicians [47].

Despite safety concerns representing a significant reason for non-vaccination [48], they were not dealt with appropriately from a legal point of view.

## 6. Proposed Changes to the Vaccination Campaign

Comparing the vaccine options of Romania to those of other nations may aid in preparation for future occurrences.

In certain nations, such as Austria, vaccination was mandated by political choices. Others, such as Bulgaria, France, Germany, and the Czech Republic, imposed vaccinations for various demographic groups [49]. Italy was the first to mandate vaccinations for healthcare workers by the end of 2021, and for all people over 50 by the start of 2022 [50]. Other nations have chosen to respect patients’ free will by requiring informed consent, although still imposing restrictions on those who refuse vaccination.

While contemplating mandating vaccination to increase the vaccination rate, it is essential to assess the Israeli experience, suggesting that less public pressure and a more significant commitment to ensuring informed consent are desirable [12]. Individual freedom of choice is a powerful argument for those who oppose obligatory vaccination [51], and authorities should consider that mandating vaccination could result in a reluctance to be vaccinated [52]. Ultimately, it is a political decision, but the state must better prepare the legal framework.

The process was also challenging for other European nations. For instance, although Ireland was one of the countries where vaccination was successful, legal unpreparedness regarding informed consent for impaired populations and children and young people was alleged [53].

Future vaccination campaigns should meet all legal considerations while serving the community’s best interests. Unlike ethical dilemmas, legal duties are obligatory. The authorities should decide ab initio to annotate Romanian laws and revise the informed consent form or to continue using the existing document.

In addition, we proposed establishing step D (Figure 2), wherein patients might interact with a physician. Although all documents could be delivered at the registration desk, the receptionist must instruct patients to complete only the triage questionnaires. Every patient should be able to speak with a physician and receive the necessary vaccination information. Research on optimizing patient flow should be conducted to assess the best model for vaccination centers while complying with the relevant laws.

Political decisions should also be taken regarding vaccine injury compensation. There are a variety of systems regarding compensation for vaccine injury, varying from the total to “no-fault” compensation systems. Since the legal framework created the physicians’ liability, Romanian authorities should decide on a national compensatory scheme to ease physicians’ burdens.

## 7. Conclusions

From a legal standpoint, we argue that the vaccination effort focused mainly on public health and, to a lesser extent, on individual rights. Inadequate legal preparation led to medical law infringement. Media and online sources replaced the lawful process of informing the patient in Romania. Now, physicians might face liability for failing to respect patients’ informed consent obligations. In certain instances, criminal liability may be asserted in addition to civil liability. Additionally, many physicians could lose their insurance coverage and would be unprotected against future claims.

As soon as the COVID-19 pandemic emerged, society called on physicians to be “heroes” on the “front lines.” The authorities issued recommendations and decisions to be implemented by the medical personnel but failed to adjust the legal framework to the new medical context. Consequently, physicians are at significant risk of being sued.

This learned lesson is relevant given the likelihood of future pandemics or other health crises. Building confidence and overcoming widespread mistrust are, in our view, inextricably related to strict adherence to medical legislation established to safeguard patients’ autonomy and freedom of choice.

## Figures and Tables

**Figure 1 vaccines-10-01871-f001:**
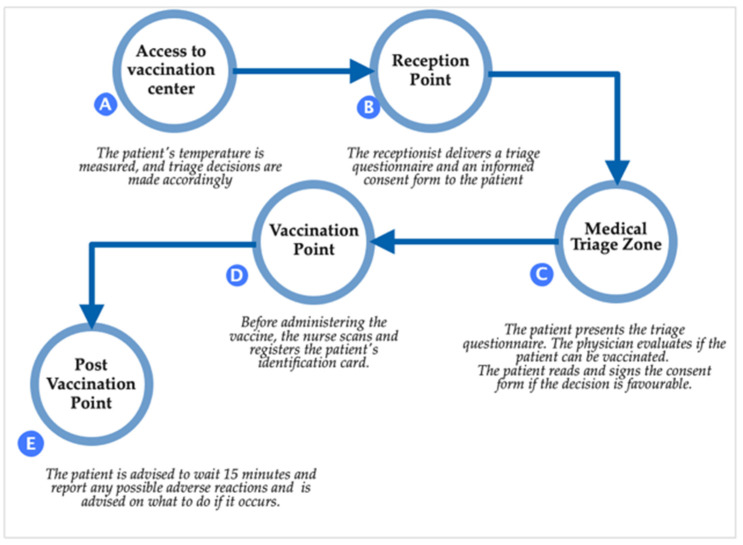
Patient traffic flow.

**Figure 2 vaccines-10-01871-f002:**
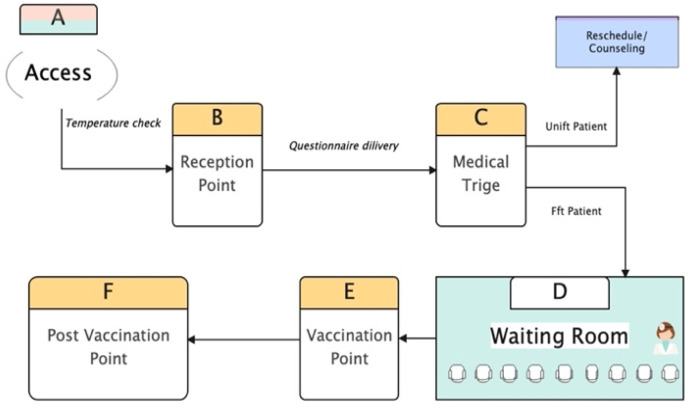
Proposed patient traffic flow.

**Table 1 vaccines-10-01871-t001:** Mandatory legal elements of patient’s informed consent.

Mandatory Legal Elements	Legal Reference
1. Information regarding certain aspects related to the medical act:	article 660, Law no. 95/2006 and article 6, Law no. 46/2003
- diagnosis - medical act’s nature and purpose - potential risks and consequences of treatment - viable treatment alternatives - risks and consequences of treatment alternatives - prognosis of the disease without treatment	
2. The identity and professional status of health service providers3. Rules and customs that must be respected by patients during hospitalization	article 5, Law no. 46/2003
4. Available medical services and how to use them	article 4, Law no. 46/2003
5. Patient’s consent is mandatory for the collection, storage, use of all biological products taken from his/her body, in order to establish the diagnosis or the treatment with which he/she agrees.	article 18, Law no. 46/2003
6. Patient’s consent is mandatory in case of his/her participation in clinical medical education and scientific research.	article 19, Law no. 46/2003

**Table 2 vaccines-10-01871-t002:** Mandatory requirements for the informed consent process.

The information must be presented using a scientific level reasonable for the patient’s understanding
2. The patient must be informed that he/she is entitled to a second medical opinion
3. The patient declares if he/she wants to be further informed regarding his/her health status
4. If patients who assume responsibility for their decision in writing refuse medical acts, the doctor must explain the consequences of refusing or ceasing medical acts.

Legal references: law no.95/2006, article 660 and law no.46/2003, articles 7, 11 and 13.

**Table 3 vaccines-10-01871-t003:** Exclusion clauses in insurance agreements in Romania.

Insurance Clause
“The insurer does not grant compensation for claims resulting from damages caused because the medical assistance of the injured party or the deceased was provided without the patient’s informed consent, except for some cases.”
2. “The insurer does not grant compensation for damages caused by medical services of which the patient or his relatives were not informed and for which they did not consent, except in cases of maximum emergency, when in the absence of immediate intervention, the patient’s life would be endangered.”
3. “The insurer does not grant compensation in case of claims for material damages resulting from medical assistance provided without informed consent granted according to legal regulations if the necessary conditions for granting such consent were met.”

## Data Availability

All data mentioned in the manuscript are available from the corresponding author if requested.

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
