# Peer review of "Informed Consent in Mass Vaccination against COVID-19 in Romania: Implications of Bad Management"

_vaccines, 2022, doi:10.3390/vaccines10111871_

Round 1
Reviewer 1 Report
I have read the article carefully, I think the topic is interesting and current although focused on a single nation issue. Some suggestions for improvement:
- I think the authors should better specify whether all doctors' insurances have those clauses and whether in any case it is the hospital administration that will cover any claims for compensation
- in addition to indicating that many activities were not adequate, the authors could propose a solution or at least a proposal for improvement in a schematic way
- the authors could briefly mention that the lack of valid consent in one activity has also led to judicial problems in other circumstances ex: doi: 10.1111 / vox.13106.
Author Response
Dear Academic Editor and Reviewers
Re: Informed Consent in Mass Vaccination against Covid-19 in Romania: Implications of bad management
Thank you for your careful consideration of our manuscript and insightful comments. We appreciate the thorough review and insightful suggestions. We consider that making the suggested adjustments helps improve the manuscript. Please note below our answers to your comments and let us know if we can address any further comments or provide any additional information for our manuscript to meet the Vaccines publication criteria fully.
Best regards,
Maria Cristina Plăiașu, PhD
Reviewer #1:
I have read the article carefully, I think the topic is interesting and current although focused on a single nation issue. Some suggestions for improvement:
We would like to thank reviewer #1 for his/her valuable comments and observations regarding the submitted manuscript, which helped us to improve our manuscript. We took into account all the issues raised by the reviewer, and we revised the manuscript accordingly as follows:
- I think the authors should better specify whether all doctors' insurances have those clauses and whether in any case, it is the hospital administration that will cover any claims for compensation.
Author's response: This comment was addressed in the revised version of the manuscript from line 241 to line 244 and 246 to 257.
- In addition to indicating that many activities were not adequate, the authors could propose a solution or at least a proposal for improvement in a schematic way.
Author's response: Thank you very much for this suggestion. This comment was addressed in the revised version of the manuscript from line 292 to line 398.
- The authors could briefly mention that the lack of valid consent in one activity has also led to judicial problems in other circumstances ex: DOI: 10.1111 / vox.13106.
Author's response: This comment was addressed in the revised version of the manuscript from line 232 to line 233.
Reviewer 2 Report
I was invited to revise the paper entitled "Informed Consent in Mass Vaccination against Covid-19 in Romania: Implications of bad management". The aim of this study was to evaluate the legal implication of malpractices linked to the sign of informed consent prior covid vaccination in Romania. A deep revision of Romanian procedures was performed explaining in dept all legal implications linked to the informed consent. In addition, Authors described how medical malpractice situations were handled.
The paper was well written and the aim of the study was clearly presented.
In my opinion, Authors should compare the situation of Romania with other countries, in particular with other European Countries.
In addition, point 5.2 (low vaccination rate) should be deeply described and compared to other countries. In my opinion, low vaccination coverage cannot be explained by the inadequate informed consent. Romania has low coverages also for all pediatric vaccination (one of the lowest rate for MMR vaccination in Europe), so this point should be improved.
Author Response
Dear Academic Editor and Reviewers
Re: Informed Consent in Mass Vaccination against Covid-19 in Romania: Implications of bad management
Thank you for your careful consideration of our manuscript and insightful comments. We appreciate the thorough review and insightful suggestions. We consider that making the suggested adjustments helps improve the manuscript. Please note below our answers to your comments and let us know if we can address any further comments or provide any additional information for our manuscript to meet the Vaccines publication criteria fully.
Best regards,
Maria Cristina Plăiașu, PhD
Reviewer #2:
I was invited to revise the paper entitled "Informed Consent in Mass Vaccination against Covid-19 in Romania: Implications of bad management". The aim of this study was to evaluate the legal implication of malpractices linked to the sign of informed consent prior covid vaccination in Romania. A deep revision of Romanian procedures was performed explaining in dept all legal implications linked to the informed consent. In addition, Authors described how medical malpractice situations were handled.
The paper was well written and the aim of the study was clearly presented.
We would like to thank reviewer #2 for his/her valuable comments and observations regarding the submitted manuscript, which helped us to improve our manuscript. We took into account all the issues raised by the reviewer, and we revised the manuscript accordingly as follows:
- In my opinion, Authors should compare the situation of Romania with other countries, in particular with other European Countries.
Author's response: This comment was addressed in the revised version of the manuscript from line 84 to line 89, line 171 to 174, and line 293 to line 311.
- In addition, point 5.2 (low vaccination rate) should be deeply described and compared to other countries. In my opinion, low vaccination coverage cannot be explained by the inadequate informed consent. Romania has low coverages also for all pediatric vaccination (one of the lowest rate for MMR vaccination in Europe), so this point should be improved.
Author's response: This comment was addressed in the revised version of the manuscript from line 259 to line 279.
Round 2
Reviewer 1 Report
the authors have greatly improved the text in accordance with the recommendations. I believe that the contribution is publishable
Reviewer 2 Report
Authors improved the manuscript addressing all comments raised during my revision. The paper can be accepted for publication in my opinion.